# Management of Myositis-Associated Interstitial Lung Disease

**DOI:** 10.3390/medicina57040347

**Published:** 2021-04-03

**Authors:** Tomoyuki Fujisawa

**Affiliations:** Second Division, Department of Internal Medicine, Hamamatsu University School of Medicine, 1-20-1 Handayama Higashi-ku, Hamamatsu 431-3192, Japan; fujisawa@hama-med.ac.jp; Tel.: +81-(53)-435-2263; Fax: +81-(53)-435-2354

**Keywords:** dermatomyositis, polymyositis, myositis, interstitial lung disease, treatment, myositis-specific autoantibody

## Abstract

Idiopathic inflammatory myopathies, including polymyositis (PM), dermatomyositis (DM), and clinically amyopathic DM (CADM), are a diverse group of autoimmune diseases characterized by muscular involvement and extramuscular manifestations. Interstitial lung disease (ILD) has major pulmonary involvement and is associated with increased mortality in PM/DM/CADM. The management of PM-/DM-/CADM-associated ILD (PM/DM/CADM-ILD) requires careful evaluation of the disease severity and clinical subtype, including the ILD forms (acute/subacute or chronic), because of the substantial heterogeneity of their clinical courses. Recent studies have highlighted the importance of myositis-specific autoantibodies’ status, especially anti-melanoma differentiation-associated gene 5 (MDA5) and anti-aminoacyl tRNA synthetase (ARS) antibodies, in order to evaluate the clinical phenotypes and treatment of choice for PM/DM/CADM-ILD. Because the presence of the anti-MDA5 antibody is a strong predictor of a worse prognosis, combination treatment with glucocorticoids (GCs) and calcineurin inhibitors (CNIs; tacrolimus (TAC) or cyclosporin A (CsA)) is recommended for patients with anti-MDA5 antibody-positive DM/CADM-ILD. Rapidly progressive DM/CADM-ILD with the anti-MDA5 antibody is the most intractable condition, which requires immediate combined immunosuppressive therapy with GCs, CNIs, and intravenous cyclophosphamide. Additional salvage therapies (rituximab, tofacitinib, and plasma exchange) should be considered for patients with refractory ILD. Patients with anti-ARS antibody-positive ILD respond better to GC treatment, but with frequent recurrence; thus, GCs plus immunosuppressants (TAC, CsA, azathioprine, and mycophenolate mofetil) are often needed in order to achieve favorable long-term disease control. PM/DM/CADM-ILD management is still a therapeutic challenge for clinicians, as evidence-based guidelines do not exist to help with management decisions. A few prospective clinical trials have been recently reported regarding the treatment of PM/DM/CADM-ILD. Here, the current knowledge on the pharmacologic managements of PM/DM/CADM-ILD was mainly reviewed.

## 1. Introduction

Idiopathic inflammatory myopathies (IIMs) are a diverse group of autoimmune diseases characterized by muscular involvement and extramuscular manifestations, including those in the skin and lungs [1,2]. Polymyositis (PM) and dermatomyositis (DM) are major subtypes of IIMs [1,2], and clinically amyopathic DM (CADM) is defined as the presence of a typical skin rash of classic DM with minimal or no features of muscular manifestations [3,4]. Patients with PM, DM, or CADM often present with interstitial lung disease (ILD), with a prevalence of approximately 40% [5,6]. Importantly, ILD has the most severe extramuscular involvement in IIMs, which is deeply related to a reduced quality of life and worse prognosis [7,8,9,10,11,12]. Thus, optimal management of PM-/DM-/CADM-associated ILD (PM-/DM-/CADM-ILD) is important for clinicians in real-world clinical practice.

Clinical courses and prognoses of PM-/DM-/CADM-ILD are heterogeneous. As for underlying diseases, DM-/CADM-associated ILD (DM-/CADM-ILD) is more refractory to treatment and has poorer prognosis than PM-associated ILD [8,10,13,14]. Regarding ILD forms, the acute/subacute form is defined as progressive ILD with deterioration within 3 months. The chronic form is defined as a slowly progressive ILD presenting with gradual deterioration over a period greater than 3 months, or as stable ILD without any progression for more than 3 months—the latter of which is less frequent. Patients with acute/subacute forms of PM/DM/CADM-ILD had a worse survival than those with the chronic form [10,15] (five-year survival rate: 52% in the acute/subacute form and 87% in the chronic form [10]). The acute form of ILD, also called rapidly progressive ILD, showing an acute worsening of dyspnea with a widespread alveolar abnormality on the chest radiological findings, has been recognized as a serious life-threatening condition with the poorest prognosis in PM-/DM-/CADM-ILD [11,15,16]. Recent studies have emphasized the importance of the assessment of myositis-specific autoantibodies (MSAs), which are closely linked to clinical phenotypes of myositis-associated ILD [17,18].

Evidence that supports optimal treatments for patients with PM/DM/CADM-ILD is limited. Most of the reports are retrospective nonrandomized observational studies and case series, although a few prospective therapeutic studies have recently been reported [19,20,21]. In real-world clinical practice, clinicians should consider therapeutic strategies for patients with PM-/DM-/CADM-ILD based on disease progression, clinical phenotype, and potential prognostic factors (e.g., ILD form and MSA status). This review mainly describes the existing evidence on the pharmacologic managements of PM-/DM-/CADM-ILD. 

## 2. Clinical Phenotype Associated with MSAs

MSA status is not a part of the diagnostic and classification criteria [2]; however, the presence of specific MSAs has been recognized to represent a distinct clinical phenotype in myositis and myositis-associated ILD. Approximately 80% of patients with PM-/DM-/CADM-ILD have certain types of MSAs [18,22]. In particular, the anti-aminoacyl tRNA synthetase (ARS) antibody and the anti-melanoma differentiation-associated gene 5 (MDA5) antibody are two major types of MSAs characterized by a high prevalence of PM-/DM-/CADM-ILD [17,18].

Anti-PM/Scl antibodies are one of myositis-associated autoantibodies, which are mostly found in Caucasian patients with PM/systemic scleroderma (Ssc) overlap syndrome, as well as in patients with PM or Ssc. Anti-PM/Scl antibodies are very rarely found in patients with PM-/DM-/CADM-ILD [23].

### 2.1. Anti-ARS Antibody

Anti-ARS antibodies are detected in 35–50% of patients with PM-/DM-/CADM-ILD [17,18,22]. Eight types of specific anti-ARS antibodies (Jo-1, PL-7, PL-12, EJ, OJ, KS, Zo, and Ha) have been identified. Patients with anti-ARS antibodies can present similar clinical characteristics, including myositis, ILD, polyarthritis, Raynaud’s phenomenon, and mechanic’s hands, and are known as antisynthetase syndrome (ASS) [24]. In addition, patients with different types of anti-ARS antibodies have unique features and prognoses [25,26]. The presence of anti-PL12 and anti-PL7 is associated with severe ILD, whereas the presence of anti-Jo1 is related to frequent arthritis and muscle involvement [24,27,28]. Patients with anti-OJ and anti-KS often present ILD alone [25]. Several studies have demonstrated that patients with anti-ARS antibody-positive PM-/DM-ILD respond well to glucocorticoid (GCs) treatments and have a comparatively favorable prognosis [17,22,29,30] with 90% achieving the five-year survival rate. However, GC treatment alone sometimes causes a relapse of ILD [29,30,31]. Thus, a combination therapy of GCs plus immunosuppressants is often performed in patients with anti-ARS antibody-positive PM/DM-ILD in order to achieve favorable long-term disease control.

### 2.2. Anti-MDA5 Antibody

Anti-MDA5 antibodies, previously termed anti-CADM140 antibodies, were originally identified in patients with CADM [32]. The autoantigen was later determined to be an RNA helicase encoded by the MDA5 gene, which plays a critical role in the innate immune defense against virus [33,34]. Anti-MDA5 antibodies are present in patients with classic DM with muscular involvement, as well as with CADM, but not in patients with PM [17,18,35,36]. The anti-MDA5 antibody is associated with a high incidence of ILD in DM/CADM patients [34], especially rapidly progressive ILD with respiratory failure and fatal outcomes [37,38,39,40].

The prevalence of anti-MDA5 antibodies is 30–50% in Asian patients with DM-/CADM-ILD [17,18]. The presence of anti-MDA5 antibody is associated with increased 90-day mortality [41] and has been recognized as a strong predictor of poor prognosis in patients with PM-/DM-/CADM-ILD [17,37,38]. The one-year survival rate is 60–70% in patients with anti-MDA5 antibody-positive DM-/CADM-ILD. Importantly, the majority of fatal outcomes in patients with anti-MDA5 antibody-positive DM-/CADM-ILD occurred within the first 6 months after the initial diagnosis [17,38,40]. Because anti-MDA5 antibody-positive rapidly progressive ILD is the most severe phenotype in PM-/DM-/CADM-ILD with the poorest prognoses, a combined immunosuppressive regimen, including high-dose GCs, oral calcineurin inhibitors (CNIs), and intravenous cyclophosphamide (IVCY) pulse, is recommended [18,20,42]. 

The pathogenesis of anti-MDA5 antibody-positive DM/CADM-ILD is not yet fully understood; however, involvement of the macrophage activation has been recently reported [40,43,44,45]. In fatal cases of rapidly progressive ILD with anti-MDA5 antibody-positive CADM, systemic activation of the macrophages is observed in many organs [44]. The serum ferritin level is a marker of the systemic activation of the macrophages, and hyperferritinemia is often found in rapidly progressive ILD with anti-MDA5 antibodies [34,37,43]. In addition, a high serum ferritin level is associated with poor prognosis in anti-MDA5 antibody-positive DM-/CADM-ILD [40,46].

## 3. Treatment

Although no consensus guidelines exist for the management of myositis-associated ILD, GCs or GCs plus immunosuppressants are considered to be the cornerstone of treatments for PM-/DM-/CADM-ILD [8,9,11,47]. Cyclophosphamide (CY), azathioprine (AZA), mycophenolate mofetil (MMF), CNIs (tacrolimus (TAC) and cyclosporin A (CsA)), and rituximab (RTX) are used in combination with corticosteroid for the treatment of PM-/DM-/CADM-ILD [47,48,49,50,51,52], which mostly relies on uncontrolled retrospective analyses or case series. Recently, a few prospective trials have reported the therapeutic efficacy of GCs plus CNIs for patients with PM-/DM-/CADM-ILD [19,21]. Table 1 presents a list of immunosuppressive agents used in the treatment of PM-/DM-/CADM-ILD.

### 3.1. Glucocorticoids

GCs (e.g., prednisolone (PSL)) are still the mainstay of treatments for PM-/DM-/CADM-ILD; this is based on historical precedent and retrospective case series [9,10,47,53,54,71,72]. Patients with acute/subacute ILD are often resistant to initial GC monotherapy, and thus combination therapy with GCs and immunosuppressants is highly recommended as an initial treatment [48,51,54,59,73]. Given the necessity of long-term treatments, poor tolerability of long-term use of GCs, and the recurrence of ILD to GCs monotherapy [29,30,50], GCs plus immunosuppressants for patients with chronic forms of PM-/DM-/CADM-ILD are commonly used. Overall, previous studies have demonstrated the therapeutic advantage of the early administration of immunosuppressants to GCs as an initial treatment for PM-/DM-ILD [51,55]. Oral GCs are used for patients with chronic ILD, whereas patients with acute/subacute ILD are typically treated with intravenous pulse therapy of methylprednisolone (mPSL; 1000 mg/day for 3 days) and subsequent high doses of oral PSL (0.75–1 mg/kg/day) [10,47,74,75].

### 3.2. Calcineurin Inhibitors

TAC and CsA are two members of CNIs that target T-cell activation. Recent studies have shown the efficacy of CsA and TAC even in severe forms of PM-/DM-/CADM-ILD [51,52,57,58,73,76,77,78,79]. The use of CNIs is becoming more common in the treatment of PM-/DM-/CADM-ILD, especially in East Asian countries.

#### 3.2.1. Cyclosporine A

Early intervention with CsA plus GCs has been found to improve the prognosis of DM-/CADM-ILD [51,56,57,73,80]. Go et al. retrospectively reported that patients with DM-/CADM-ILD who received CsA treatment within 2 weeks of diagnosis (early treatment group) had a better disease stabilization and survival rate compared with those who received CsA after the trial of other immunosuppressants (delayed treatment group) [51]. Cavagna et al. described the long-term efficacy and safety of CsA in patients with GCs-refractory ILD with anti-Jo1 [57]. They found that adding CsA was effective at improving pulmonary functions and reducing the GC dose with tolerable safety profiles. In a systematic review, Barba et al. reported that the survival rate after 3 months of rapidly progressive ILD was 51.7% and 69.2% for GCs alone and CsA, respectively [72]. The starting dose of CsA is 3 mg/kg/day (twice daily), and common adverse effects of CsA include renal dysfunction, hypertension, and gastrointestinal disorder. Monitoring and maintaining a whole-blood trough level of 100–150 ng/mL is suggested in order to avoid adverse events [21,56,57].

#### 3.2.2. Tacrolimus

TAC is a second-generation CNI with a 100-fold great potency for the inhibition of T-cell activation compared with CsA. Recent studies have demonstrated that the administration of TAC is effective in patients with PM-/DM-/CADM-ILD, even in those resistant to CsA or CY regimens [52,58,76]. Kurita et al. retrospectively showed that adding TAC to conventional therapy (GCs alone or GCs plus other immunosuppressants, except for TAC) improves the event-free survival compared with conventional therapy in patients with PM-/DM-ILD [52]. Sharma et al. assessed a cohort of patients with PM-/DM-ILD from the USA, and reported that 72% of patients (17 of 18 patients) who failed to respond to conventional treatments (prednisone plus AZA, methotrexate, or MMF) showed improvement in myositis and ILD when receiving TAC [58]. A systematic review by Ge and colleagues demonstrated that TAC improves both muscle strength and pulmonary function in patients with PM-/DM-ILD, with tolerable adverse events [81]. In real-world clinical practice, a TAC dose of 0.075 mg/kg/day is administered twice daily. Common adverse events of TAC are similar to those of CsA, including renal dysfunction, hypertension, hypomagnesemia, and tremors [81]. Monitoring and maintaining a whole-blood trough level of 5–10 ng/mL is usually recommended [21]. Takada et al. recently conducted a single-arm prospective clinical trial to evaluate the efficacy of combination treatment with PSL and TAC in patients with PM-/DM-ILD. They reported that PSL plus TAC treatment achieved 88% for the 52-week survival rate and 76.4% for the 52-week progression-free survival (PFS) rate, suggesting that initial treatment with PSL plus TAC is effective in improving short-term mortality [19]. 

GCs plus CNIs are promising for the initial treatment of PM-/DM-/CADM-ILD; however, which CNI regimens, TAC or CsA, are more effective remains unknown. Recently, a prospective, open-label, randomized trial was recently to compare the efficacy between treatments with PSL plus TAC and PSL plus CsA in 58 patients with PM-/DM-/CADM-ILD [21]. The PFS rates at 52 weeks were 87% and 71% in the TAC and CsA regimens, respectively, which suggests that PSL plus TAC treatment may be superior to PSL plus CsA treatment for achieving a higher PFS rate in PM-/DM-/CADM-ILD patients. No significant difference was observed in the 52-week survival rate between the TAC (97%) and CsA (93%) regimens. These findings suggest that treatment with PSL and CNIs is a promising initial therapy for PM-/DM-/CADM-ILD patients.

### 3.3. Other Immunosuppressants

Several immunosuppressants, except for CNIs, including CY, AZA, and MMF, have been historically used for PM-/DM-/CADM-ILD treatment. However, no prospective study has been performed to compare the efficacy among these immunosuppressants. Mira-Avendano et al. retrospectively analyzed the effectiveness of CY, AZA, and MMF for patients with GC-resistant PM-/DM-ILD. They found that each of these immunosuppressants was equally associated with the stabilization of pulmonary functions and the improvement of dyspnea [49].

#### 3.3.1. Cyclophosphamide

CY has been used mainly for patients with severe or refractory PM-/DM-/CADM-ILD [5,48,59,82,83,84]. Because of its toxicity profile, including the risk of secondary cancer and gonadal dysfunction, CY administration should be considered for most aggressive or refractory forms of ILD. IVCY (e.g., 300–800 mg/m^2^ every 4 weeks [48]) is preferred over oral CY in order to reduce the cumulative dose and the risk of adverse effects.

Yamasaki et al. reported the efficacy of IVCY in patients with progressive PM-/DM-/CADM-ILD in an open-label trial [48]. Seventeen patients with PM-/DM-/CADM-ILD received IVCY (300–800 mg/m^2^, at least 6 times every 4 weeks) together with oral PSL (0.5–1 mg/kg/day), and responses to the treatment were evaluated. Of the 17 patients, 11 had improved dyspnea, 8 had improved vital capacity, and 9 showed improved high-resolution computed tomography (HRCT) findings. A systematic review of the 12 studies summarized the efficacy and adverse effects of CY use in the management of IIM and IIM-associated ILD [84]. According to the analyses, 57.6% (34 of 59) and 67.3% (35 of 52) of patients improvements in their forced vital capacity and HRCT scores, respectively. Regarding acute/subacute ILD, 58.1% (25 of 43) of patients survived after IVCY treatment. The efficacy of combination therapy of PSL, IVCY, and CsA for rapidly progressive DM-ILD was previously evaluated in a prospective pilot trial [59]. Ten patients were treated with a high-dose of PSL, 10–30 mg/kg of IVCY every 3–4 weeks, and 2–4 mg/kg/day of CsA. Half of the patients had a favorable outcome of more than 2 years, although five patients died of respiratory failure within 3 months.

Patients with DM-/CADM-ILD with anti-MDA5 antibodies are associated with a high incidence of rapidly progressive ILD with fatal outcomes [33,34,37]; however, the standard treatments for patients with anti-MDA5 antibody-positive DM-/CADM-ILD have not been established yet. Tsuji et al. recently reported a single-arm clinical trial to evaluate the efficacy and safety of the combined immunosuppressive regimen with GCs, TAC, and IVCY (500–1000 mg/m^2^ per 2 weeks, six times, every 4–8 weeks for a total of 10–15 times) in patients with anti-MDA5 antibody-positive DM-/CADM-ILD. In this study, the TAC was adjusted to retain higher trough levels (10–12 ng/mL). They showed that the 12-month survival rates improved in the immunosuppressive regimen group (85%) compared with those in the historical control group (33%), who were treated with conventional step-up treatment (high-dose GCs and stepwise addition of immunosuppressant based on the deterioration of clinical conditions) [20]. Opportunistic infections including cytomegalovirus reactivation and pneumocystis pneumonia are major clinical concerns for the combined immunosuppressive therapy with GCs, CNIs, and IVCY [20,85], which clinicians should be aware of. 

#### 3.3.2. Antimetabolites (AZA and MMF)

AZA and MMF are members of the family of antimetabolite drugs, both of which are widely used for the treatment of connective tissue disease (CTD)-associated ILD because of their favorable safety profile [60,61]. Although AZA has been historically used as steroid-sparing medication for maintenance therapy after induction with CY [83], control studies supporting the use of AZA for PM-/DM-/CADM-ILD treatment are relatively few. Mira-Avendano et al., in a retrospective study, found that 13 patients with PM-/DM-ILD treated with GCs plus AZA had an improvement in dyspnea and stabilization of the pulmonary function for 12 months with a successful reduction of the PSL dose [49]. AZA is administered orally at a dose of 1–2 mg/kg, and its potential side effects include leukopenia, elevated transaminase, and opportunistic infection [50,60].

Several case series have demonstrated the therapeutic benefit of MMF for patients with PM-/DM-ILD [49,50,61,86,87]. Thirty-two patients with PM-/DM-ILD treated with MMF (2000–3000 mg/day) with prednisone had significant improvements in their % forced vital capacity (FVC) after 156 weeks [61]. MMF is relatively well tolerated, and the common adverse events of MMF include gastrointestinal intolerance, cytopenia, elevated transaminase, and infection.

Recently, a retrospective observational study was conducted to compare the long-term therapeutic effects of AZA and MMF on lung function and GC dose in 110 patients with myositis-associated ILD [50]. In this study, 66 and 44 patients were treated with AZA and MMF, respectively. In the AZA group, 1 (1.5%) and 56 patients (85%) were positive for the anti-MDA5 antibody and anti-ARS antibody, respectively. In the MMF group, 14 (32%) and 7 patients (16%) were positive for the anti-ARS antibody and anti-MDA5 antibody, respectively. Both treatment groups had an improvement in the %FVC and reduction of prednisone over 2–5 years. The AZA group had an improvement of % diffusing capacity for carbon monoxide (DLCO), and the MMF group reached stabilization of %DLCO over 5 years. Adverse events were more frequent in the AZA group (33.3%) compared with those in the MMF group (13.6%). The most frequent adverse event was transaminitis (10 (15.2%) and 1 (2.3%) patients in the AZA and MMF groups, respectively). 

### 3.4. Rituximab

RTX is a chimeric anti-CD20 monoclonal antibody that targets B cells and results in B-cell depletion. Several studies have shown the potential effects of RTX in the PM-/DM-/CADM-ILD treatment [62,63,88,89,90], especially in patients with anti-ARS antibodies. A retrospective analysis of seven patients with ASS-associated ILD (ASS-ILD) refractory to first-line therapy showed that RTX improved clinical manifestations, pulmonary functions (FVC and DLCO), and the HRCT image after 1 year [88]. Another retrospective study including 24 patients with severe ASS-ILD treated with RTX and with a median follow-up time of 52 months demonstrated that RTX treatment significantly improved both the pulmonary function tests and HRCT images [62]. In this study, seven patients with acute ILD responded with FVC and DLCO of more than 30%. Doyle et al. reviewed 25 patients with recurrent or progressive ASS-ILD (16 patients with Jo1, 6 with PL-12, and 3 with PL-7) who were treated with RTX [63]. They demonstrated that with RTX treatment, most of the patients achieved stability or improvement in CT images and pulmonary function (e.g., FVC and DLCO) after 1 and 3 years of follow-up.

Additionally, recent retrospective studies have also demonstrated the effectiveness of RTX in patients with anti-MDA5 antibodies [64,65,91,92,93]. A case series of four patients with amyopathic DM (ADM) with rapidly progressive ILD who failed to respond to high-dose GCs and immunosuppressants reported that RTX treatments improved or stabilized respiratory symptoms, pulmonary function tests, and HRCT images in all patients after 6 months to 2 years of follow-up [64]. A retrospective review of 11 patients with anti-MDA5 antibody-positive DM-ILD (including eight patients with rapidly progressive ILD) revealed that eight patients (73%) have improved lung function and HRCT after RTX treatment [65]. Infection episodes occurred in 45% of the patients, which are major adverse events of RTX.

According to these findings, RTX may be a candidate of salvage therapy in the management of refractory cases and of rapidly progressive cases in PM-/DM-/CADM-ILD. Moreover, a prospective randomized control trial comparing the efficacy of RTX and CY as a first-line treatment in CTD-associated ILD (RECITAL study) has been performed [94]. 

### 3.5. Tofacitinib

Tofacitinib (TOF) is a Janus kinase inhibitor that blocks multiple cytokine signaling, including interferon and IL-6. Rapidly progressive DM-/CADM-ILD is characterized by the overproduction of multiple cytokines, especially in patients with anti-MDA5 antibodies [95,96]. Emerging data support the use of TOF in patients with DM-/CADM-ILD with anti-MDA5 antibodies [66,67,68,97,98]. Kurasawa et al. retrospectively reviewed the efficacy of TOF (10 mg/day) for patients with anti-MDA5 antibody-positive DM-ILD who were resistant to triple therapy, including high-dose GCs, CsA, and CY [66]. They reported that 60% (3 of 5) of patients treated with TOF had a good response and survived, whereas none of the six historical controls without TOF treatment, with a similar poor prognostic factor, survived, suggesting that TOF treatment may be a potential candidate for refractory DM-ILD with anti-MDA5 antibodies. The major adverse events of TOF were infection, especially viral infection (cytomegalovirus reactivation in 100% and herpes zoster infection in 60% of patients receiving TOF treatment). TOF efficacy in patients with early-stage ADM-associated ILD (ADM-ILD) with anti-MDA5 antibodies has been evaluated in a single-arm open-label trial [68]. In this study, 18 patients with anti-MDA5 antibody-positive ADM-ILD were prospectively treated with GCs and TOF (10 mg/day), and the survival of the patients with TOF treatment was compared with that of a historical control who received conventional treatment. The survival rate at 6 months was significantly higher in the TOF group (100%) than that in the historical control (78%).

### 3.6. Intravenous Immunoglobulin

The treatment of intravenous immunoglobulin (IVIG) in combination with GCs has been shown to be effective for patients with refractory myositis [99,100,101]. However, data supporting its use for PM-/DM-/CADM-ILD treatment are relatively limited [69,70,102,103]. Therefore, routine use of IVIG is currently not recommended for PM-/DM-/CADM-ILD treatment. A case series of IVIG use (0.4 g/kg/day, for five consecutive days) in five patients with refractory PM-/DM-/CADM-ILD resistant to high-dose GCs, CsA, and/or CY has been reported [69]. Among the five patients treated with IVIG, two survived with stabilized ILD, and three died of respiratory failure. No serious adverse events were observed. Common side effects include headache, fever, and chills. Serious adverse events are less frequent.

### 3.7. Plasma Exchange

Plasma exchange (PE), also known as plasmapheresis, is used in the treatment of various autoimmune diseases, to remove pathogenic substances (e.g., circulating antibodies, cytokines, and immune complexes). Miller et al. showed no clinical benefit of PE in the controlled trial comparing PE with leukapheresis to sham apheresis in 39 patients with PM/DM without ILD [104].

Recent publications have demonstrated the expectation of PE as a salvage therapy for rapidly progressive anti-MDA5 antibody-positive DM-/CADM-ILD [105,106,107]. Abe et al. reported a retrospective review of 10 patients with rapidly progressive anti-MDA5 antibody-positive DM-/CADM-ILD refractory with the initial treatment of intensive immunosuppressive therapy (high-dose GCs, CNIs, and IVCY) [105]. Of the 10 patients, six received PE and four did not. The survival rate after 1 year was significantly higher in patients with PE (100%) than those without PE (25%). Shirakashi et al. recently reported the efficacy of PE as an effective adjuvant treatment for anti-MDA5 antibody-positive DM-/CADM-ILD [107]. Thirty-eight patients with anti-MDA5 antibody-positive DM-/CADM-ILD treated with intensive immunosuppressive therapy (high-dose GCs, CNI, and IVCY) were retrospectively reviewed. Of the 38 patients, 13 had the progression of hypoxia without a response to the immunosuppressive therapy. Of the 13 patients refractory to the immunosuppressive therapy, eight received PE and five of them (63%) survived; however, five patients without PE died. In this study, PE was performed one to three times per week, for 3 to 15 consecutive weeks. In PE, 1.0–1.3 volumes of plasma per session were replaced with equivalent volumes of fresh frozen plasma or 5% albumin.

## 4. Proposal of Treatment Algorithm

The clinical course in patients with PM-/DM-/CADM-ILD is quite heterogeneous. So far, large control studies to confirm optimal treatments in patients with PM-/DM-/CADM-ILD are limited. Additionally, no evidence-based guidelines are available to help with the management decision in myositis-associated ILD. Given the high mortality and difficulty of treatment of choice in PM-/DM-/CADM-ILD, referral to the specialty centers should always be considered in real-world clinical practice. Recently, a guide for the diagnosis and treatment of CTD-associated ILD has been published by the Japanese Respiratory Society (JRS) and the Japan College of Rheumatology (JCR) (currently available in Japanese only). A treatment algorithm for patients with PM-/DM-/CADM-ILD has been proposed on the basis of our experiences, a review of the literature, and the guide by the JRS and JCR in Figure 1.

The initial treatment should be determined by considering the severity of ILD (e.g., clinical symptoms, pulmonary functions, and chest HRCT findings) and factors associated with poor prognoses, as shown in Table 2 (e.g., acute/subacute form, rapidly progressive ILD, anti-MDA5 antibody-positive, old age, hypoxia, elevated ferritin, elevated Krebs von den Lungen-6 (KL-6), elevated c-reactive protein (CRP), and low FVC [10,22,45,108]). Among them, ILD forms (acute/subacute or chronic) and clinical phenotypes based on MSA status (anti-MDA5 antibody-positive or anti-ARS antibody-positive) are important distinctions to determine the initial treatments for PM-/DM-/CADM-ILD.

Patients with acute/subacute ILD should be considered in order to provide prompt and aggressive treatments with high-dose PSL (preceded by mPSL pulse therapy in severe cases) combined with CNIs (TAC or CsA), regardless of MSA status, because of the possibility of a poor outcome. Importantly, patients with rapidly progressive ILD have the poorest outcome and should be treated with a more aggressive triple combination therapy of high-dose PSL, CNIs, and IVCY as soon as possible. Clinicians should pay attention to opportunistic infections during triple combination therapy. When patients with subacute ILD are positive for anti-MDA5 antibodies, triple combination therapy of high-dose PSL, CNIs, and IVCY can be considered, especially when multiple prognostic factors exist (e.g., old age, hypoxia, elevated ferritin, elevated KL-6, elevated CRP, and low FVC). Treatment with high-dose PSL and CNIs may be selected for patients with anti-MDA5 antibody-positive subacute ILD without any prognostic factors. For patients with acute/subacute ILD who are negative for anti-MDA5 antibody, including patients with anti-ARS antibody, treatment with PSL and CNIs should be continued. In patients with refractory disease despite adequate treatments including triple combination therapy of PSL, CNIs, and IVCY, salvage therapies (RTX, TOF, and PE) can be considered. Patients with refractory rapidly progressive ILD should also be referred to a lung transplantation center.

In patients with chronic forms of ILD, clinicians should pay attention to the chronic disease progression, including the deterioration of clinical symptoms (dyspnea and cough), pulmonary functions, and chest radiologic images. Patients with chronic progressive ILD should be treated with PSL plus immunosuppressants (e.g., TAC, CsA, MMF, and AZA). Treatment with PSL and CNIs should be considered for patients with anti-MDA5 antibody-positive chronic ILD. Given that patients with anti-ARS antibody-positive PM-/DM-/CADM-ILD have a better response to GC treatments with a favorable prognosis but frequent recurrence of ILD during the tapering of GC treatment, combination therapy of GCs and immunosuppressants (TAC, CsA, MMF, and AZA) is currently recommended. In chronic ILD without any types of MSAs, PSL plus immunosuppressant or PSL alone may be candidates for the initial treatment. If ILD is stable without any progression for more than 3 months, careful follow-up may be one of the options for management.

## 5. Conclusions

Because ILD is a leading cause of morbidity in PM/DM/CADM, ILD management in patients with PM/DM/CADM requires careful evaluation of the disease severity and prognostic factors, including ILD forms, as well as a close follow-up of patients. Recent studies have highlighted the importance of MSA status, especially anti-MDA5 and anti-ARS antibodies, for the evaluation of clinical phenotypes and for the choice of initial treatment in PM-/DM-/CADM-ILD. Combination therapy with GCs and CNIs is recommended for patients with anti-MDA5 antibody-positive DM-/CADM-ILD, because of a potential poor outcome. Rapidly progressive DM-/CADM-ILD with the anti-MDA5 antibody is the most intractable condition and requires immediate combined immunosuppressive therapy with GCs, CNIs, and IVCY. Patients with anti-ARS antibody-positive ILD respond better to GC treatment, but with frequent recurrence. Thus, treatment with GCs and immunosuppressants is often needed in order to achieve favorable long-term disease control. Randomized controlled trials are required in order to resolve clinical questions such as which immunosuppressant is the most suitable for initial induction therapy or long-term maintenance therapy in patients with PM-/DM-/CADM-ILD. In addition, prospective clinical trials based on the stratification of MSA status, especially those presenting with anti-MDA5 antibody-positive or anti-ARS antibody-positive, is needed in order to elucidate evidence-based treatment approaches in various phenotypes of myositis-associated-ILD. The other key question is which salvage therapies should be considered for refractory cases of rapidly progressive ILD. Further investigation into the pathophysiology of diseases and the identification of therapeutic targets are also important in order to develop better therapeutic strategies for patients with PM-/DM-/CADM-ILD.

## Figures and Tables

**Figure 1 medicina-57-00347-f001:**
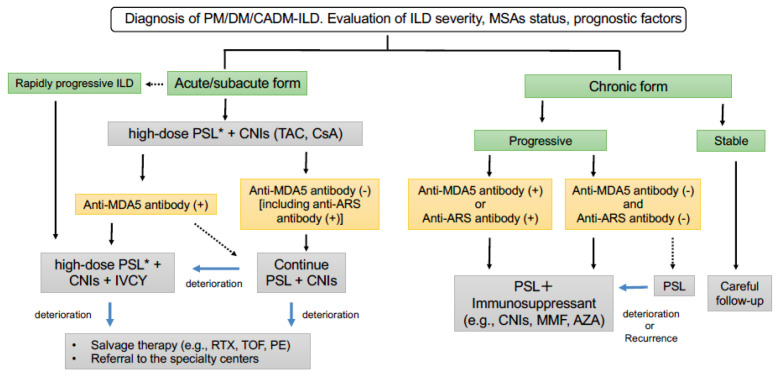
Proposal of a treatment algorithm for patients with PM-/DM-/CADM-ILD. * Methylprednisolone pulse therapy (1000 mg/day intravenous for 3 days) should be considered for severe ILD. PM—polymyositis; DM—dermatomyositis; CADM—clinically amyopathic DM; ILD—interstitial lung diseases; MSAs—myositis-specific autoantibodies; PSL—prednisolone; CNIs—calcineurin inhibitors; TAC—tacrolimus; CsA—cyclosporin A; MDA5—melanoma differentiation-associated gene 5; ARS—aminoacyl tRNA synthetase; IVCY—intravenous cyclophosphamide; MMF—mycophenolate mofetil; AZA—azathioprine; RTX—rituximab; TOF—tofacitinib; PE—plasma exchange.

**Table 1 medicina-57-00347-t001:** Immunosuppressive agents used in the treatment of PM-/CM-/CADM-associated interstitial lung diseases.

Medication	Dose	Adverse Effect	Evidence Level
Corticosteroids	Prednisolone, 0.75–1 mg/kg/day po Methylprednisolone, 1000 mg/day IV for 3 days for RP-ILD	Weight gain, hyperglycemia, hypertension, osteoporosis, and infection	Case series and retrospective studies [8,53,54,55]
Calcineurin inhibitor			
Cyclosporine A	3 mg/kg/day p.o. (adjust for trough level 100–150 ng/mL)	Renal dysfunction, hypertension, hyperkalemia, hyperglycemia, hypomagnesemia, and infection	Case series and retrospective studies [51,56,57]
Tacrolimus	0.075 mg/kg/day p.o. (adjust for trough level 5–10 ng/mL)		Case series and retrospective studies [52,58], and prospective studies [19,21]
Cyclophosphamide	300–1000 mg/m^2^, monthly pulsed IV	Nausea, leukopenia, secondary cancer, and gonadal dysfunction	Case series and retrospective studies [5,48], and prospective pilot studies [20,59]
Azathioprine	2 mg/kg/day p.o.	Leukopenia, elevated transaminase, and opportunistic infection	Case series and retrospective studies [49,50,60]
Mycophenolate mofetil	1000–3000 mg/day p.o.	Gastrointestinal intolerance, cytopenia, elevated transaminase, and infection	Case series and retrospective studies [49,50,61]
Rituximab	1000 mg IV at day 0, 14 375 mg/m^2^ IV weekly, 4 times	Infection and infusion reaction	Case series and retrospective studies [62,63,64,65]
Tofacitinib	10 mg/day p.o.	Infection (tuberculosis, bacterial, invasive fungal, viral, other opportunistic infection), leukocytopenia, and liver dysfunction	Case series and retrospective studies [66,67], and prospective pilot studies [68]
Intravenous immunoglobulin	0.4 g/kg/day for 5 consecutive days	Headache, fever, fatigue, and chills	Case series [69,70]

IV—intravenous; p.o.—per os (orally); RP—rapidly progressive.

**Table 2 medicina-57-00347-t002:** Factors associated with poor prognoses in PM-/CM-/CADM-associated interstitial lung diseases.

Factors Associated with Poor Prognoses
Acute/subacute form [10,15]
Old age [10,22]
Hypoxemia [22,31]
Anti-MDA5 antibody positive [22,30,33,37,38,41]
Elevated ferritin [17,40,41,43,109]
Elevated KL-6 [40,110]
Elevated CRP [22,110]
Low FVC [10,17]

MDA5—anti-melanoma differentiation-associated gene 5; KL-6—Krebs von den Lungen-6; CRP—c-reactive protein; FVC—forced vital capacity.

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
