# Peer review of "Management of Myositis-Associated Interstitial Lung Disease"

_medicina, 2021, doi:10.3390/medicina57040347_

Round 1

Reviewer 1 Report

I read with great interest the manuscript of Fujisawa T.

The paper is well written, complete, sober. It is also embellished with an interesting and reasonable therapeutic proposal.
I am really satisfied, I congratulate the author and I have no significant changes to propose. 

Author Response

I thank the reviewer for careful reading of our manuscript and giving useful comments. In response to the reviewer's comments, I have revised the manuscript.

My responses to the comments are as follows: 

Comments:  The paper is well written, complete, sober. It is also embellished with an interesting and reasonable therapeutic proposal. I am really satisfied, I congratulate the author and I have no significant changes to propose. 

Response:  I really appreciate the comments. Thank you so much for reviewing the manuscript.

Reviewer 2 Report

Management of Myositis-Associated Interstitial Lung Disease 

The author should be commended on a very well written and clearly laid out update which will prove very useful for the myositis community and physicians treating this difficult to treat disease. There are important updates in the medical literature that have been referred to.

Line 63 - some mention should be made to PM-Scl+ disease, even if brief

Line 73 - change ZO to Zo

Line 145 - remove Basically

Can you specify the target trough dose for maintenance - is it also 3mg/kg?

Line 166 - also worth mentioning here that the Tsuji study aimed for a higher level of 10-15ng/ml

Line 190 - change from Cyclophosphamides to Cyclophosphamide

Line 221 - ?also worth mentioning PJP in addition to CMV. Any advice about evidence for monitoring CMV titer?

Line 312 - worth mentioning in brief about the regimes of plasma exchange used in the studies. Also to mention the seminal negative study of Miller et al

Figure 1

Change Carefull to Careful

No mention about ARS+ patients in the Chronic form algorithm?

Author Response

I thank the reviewer for careful reading of our manuscript and giving useful comments. In response to the reviewer's comments, I have revised the manuscript.

My responses to the comments are as follows:

Comments:  The author should be commended on a very well written and clearly laid out update which will prove very useful for the myositis community and physicians treating this difficult to treat disease. There are important updates in the medical literature that have been referred to.

Response:  I really appreciate the comments. Thank you so much for reviewing the manuscript.

<Specific comments>

C1.  Line 63 - some mention should be made to PM-Scl+ disease, even if brief.

R1.  Thank you for pointing out an important issue. As suggested by the reviewer, I have added brief descriptions about anti-PM/Scl antibody as follows (line 74-77):

Anti-PM/Scl antibodies are one of myositis-associated autoantibodies, which are mostly found in Caucasian patients with PM/systemic scleroderma (Ssc) overlap syndrome as well as patients with PM or Ssc. Anti-PM/Scl antibodies are very rarely found in patients with PM/DM/CADM-ILD [1].

[1] Muro Y, Hosono Y, Sugiura K, Ogawa Y, Mimori T, Akiyama M. Anti-PM/Scl antibodies are found in Japanese patients with various systemic autoimmune conditions besides myositis and scleroderma. Arthritis research & therapy. 2015;17:57.

C2.  Line 73 - change ZO to Zo

R2.  Thank you for pointing out. I have revised the word (line 80).

C3.  Line 145 - remove Basically

R3.  Thank you for pointing out. I have removed the word (line 177).

C4.  Can you specify the target trough dose for maintenance - is it also 3mg/kg?

R4.  Thank you for pointing out an important issue. For maintenance, we keep whole-blood trough level of 100–150 ng/mL to avoid adverse events. CsA dose of 2-3 mg/kg/day is usually enough to keep the trough level, although It depends on patient’s situation. We have described that “Monitoring and maintaining whole-blood trough level of 100–150 ng/mL is suggested to avoid adverse events” (line 179-180).

C5.  Line 166 - also worth mentioning here that the Tsuji study aimed for a higher level of 10-15ng/ml

R5. Thank you for pointing out an important issue. We have added descriptions in line 233-234 as follows:

“In this study, TAC was adjusted to retain higher trough levels (10–12 ng/mL)”.

C6.  Line 190 - change from Cyclophosphamides to Cyclophosphamide

R6.  Thank you for pointing out our mistake. We have revised it (line 274).

C7.  Line 221 - also worth mentioning PJP in addition to CMV.

R7.  Thank you for pointing out. As suggested by the reviewer, pneumocystis pneumonia is one of important opportunistic infections. We have added “pneumocystis pneumonia” as well as cytomegalovirus reactivation (line 239).

C8.  Line 312 - worth mentioning in brief about the regimes of plasma exchange used in the studies. Also, to mention the seminal negative study of Miller et al

R8.  Thank you for pointing out issues. As suggested by the reviewer, I have briefly described the regimen of PE in the study by Shirakashi et al (line 350 – 352). Also, I have cited the controlled study by Miller et al (line 334 – 336).

“In this study, PE was performed one to three times per week for 3 to 15 consecutive weeks. In PE, 1.0-1.3 volumes of plasma per session were replaced with an equivalent volumes of fresh frozen plasma or 5% albumin.”

“Miller et al. showed no clinical benefit of plasma exchange in the controlled trial comparing plasma exchange with leukapheresis to sham apheresis in 39 patients with PM/DM without ILD [2].”

[2] Miller FW, Leitman SF, Cronin ME, Hicks JE, Leff RL, Wesley R, et al. Controlled trial of plasma exchange and leukapheresis in polymyositis and dermatomyositis. The New England journal of medicine. 1992;326:1380-4.

C9.  Figure 1 Change Carefull to Careful. No mention about ARS+ patients in the Chronic form algorithm?

R9.  Thank you for pointing out mistakes in Figure 1. We have revised them.

Carefull → Careful   

Chronic form, progressive: Anti-ARS antibody (-) → Anti-ARS antibody (+)

Reviewer 3 Report

The manuscript is well written and summarizes the current treatment of myositis-associated ILD. It will benefit from an English editing service that is familiar with myositis and ILD, since some sentences did not sound right... 

Major comments: 

A paragraph on ILD pathogenesis should be added. What is the role of MDA-5 or anti-synthetase in ILD? 

Is clinical course different in ILD subtype (i.e. ARS vs. MDA5 vs. DM vs. PM?)

  • histology
  • HRCT finding
  • natural history/course, prognosis  
  • treatment response
  • etc...

IIM-ILD: is it different from systemic sclerosis-ILD in regard to pathogenesis and outcome?

A table of poor prognostic factors should be added. 

Table summarizing the treatment efficacy of each drug or drug combination using PICOR (Patients, Intervention, Comparator, Outcome, Results) style should be added. This could show which drug works best for each disease subset.  Is CNI vs antimetabolite in a certain ILD subset? This information can help to understand the proposed treatment algorithm.

Minor comments: 

"PM/Dm/CAMD" should be replaced with "IIM".

Page 1, Line 34: "also called" myositis" should be removed.

Page 2, Ln 48. "Acute" vs. "chronic" should be defined more clearly.

line 79. "better" than what?? 

Page 3: Table 1: Dose of medication 

Figure 1. "Poor response" should be defined

Author Response

I thank the reviewer for careful reading of our manuscript and giving useful comments. In response to the reviewer's comments, I have revised the manuscript.

My responses to the comments are as follows:

Major comments: 

C1.  A paragraph on ILD pathogenesis should be added. What is the role of MDA-5 or

anti-synthetase in ILD? 

R1.  Thank you for pointing out an important issue. As suggested by the reviewer, I have added descriptions in the section of “Clinical phenotype associated with MSAs” (page 2 line 83-87, page 3 line 113-120) as follows:

“In addition, patients with different types of anti-ARS antibodies have unique features and prognosis [1, 2]. Presence of anti-PL12 and anti-PL7 are associated with severe ILD, whereas presence of anti-Jo1 is related to frequent arthritis and muscle involvement [3-5]. Patients with anti-OJ and anti-KS often present ILD alone [1]. (page 2 line 83-87)”

“The pathogenesis of anti-MDA5 antibody-positive DM/CADM-ILD has not been fully understood; however, involvement of macrophage activation has been recently reported [6-9]. In fatal cases of anti-MDA5 antibody-positive CADM-related rapidly progressive ILD, systemic activation of macrophages was observed in many organs [7]. Serum ferritin level is a marker of systemic activation of macrophages, and hyperferritinemia is often found in rapidly progressive ILD with anti-MDA5 antibody [6, 10, 11] . In addition, high serum ferritin level is associated with poor prognosis in anti-MDA5 antibody-positive DM/CADM-ILD [8, 12]. (page 3 line 113-120)”

[1] Hamaguchi Y, Fujimoto M, Matsushita T, Kaji K, Komura K, Hasegawa M, et al. Common and distinct clinical features in adult patients with anti-aminoacyl-tRNA synthetase antibodies: heterogeneity within the syndrome. PloS one. 2013;8:e60442.

[2] Fujisawa T, Hozumi H, Kono M, Enomoto N, Nakamura Y, Inui N, et al. Predictive factors for long-term outcome in polymyositis/dermatomyositis-associated interstitial lung diseases. Respiratory investigation. 2017;55:130-7.

[3] Hervier B, Devilliers H, Stanciu R, Meyer A, Uzunhan Y, Masseau A, et al. Hierarchical cluster and survival analyses of antisynthetase syndrome: phenotype and outcome are correlated with anti-tRNA synthetase antibody specificity. Autoimmunity reviews. 2012;12:210-7.

[4] Marie I, Josse S, Decaux O, Dominique S, Diot E, Landron C, et al. Comparison of long-term outcome between anti-Jo1- and anti-PL7/PL12 positive patients with antisynthetase syndrome. Autoimmunity reviews. 2012;11:739-45.

[5] Pinal-Fernandez I, Casal-Dominguez M, Huapaya JA, Albayda J, Paik JJ, Johnson C, et al. A longitudinal cohort study of the anti-synthetase syndrome: increased severity of interstitial lung disease in black patients and patients with anti-PL7 and anti-PL12 autoantibodies. Rheumatology. 2017;56:999-1007.

[6] Gono T, Kawaguchi Y, Hara M, Masuda I, Katsumata Y, Shinozaki M, et al. Increased ferritin predicts development and severity of acute interstitial lung disease as a complication of dermatomyositis. Rheumatology. 2010;49:1354-60.

[7] Gono T, Miyake K, Kawaguchi Y, Kaneko H, Shinozaki M, Yamanaka H. Hyperferritinaemia and macrophage activation in a patient with interstitial lung disease with clinically amyopathic DM. Rheumatology. 2012;51:1336-8.

[8] Fujisawa T, Hozumi H, Yasui H, Suzuki Y, Karayama M, Furuhashi K, et al. Clinical Significance of Serum Chitotriosidase Level in Anti-MDA5 Antibody-positive Dermatomyositis-associated Interstitial Lung Disease. The Journal of rheumatology. 2019;46:935-42.

[9] Horiike Y, Suzuki Y, Fujisawa T, Yasui H, Karayama M, Hozumi H, et al. Successful classification of macrophage-mannose receptor CD206 in severity of anti-MDA5 antibody positive dermatomyositis associated ILD. Rheumatology. 2019;58:2143-52.

[10] Gono T, Kawaguchi Y, Satoh T, Kuwana M, Katsumata Y, Takagi K, et al. Clinical manifestation and prognostic factor in anti-melanoma differentiation-associated gene 5 antibody-associated interstitial lung disease as a complication of dermatomyositis. Rheumatology. 2010;49:1713-9.

[11] Nakashima R, Imura Y, Kobayashi S, Yukawa N, Yoshifuji H, Nojima T, et al. The RIG-I-like receptor IFIH1/MDA5 is a dermatomyositis-specific autoantigen identified by the anti-CADM-140 antibody. Rheumatology. 2010;49:433-40.

[12] Gono T, Kawaguchi Y, Ozeki E, Ota Y, Satoh T, Kuwana M, et al. Serum ferritin correlates with activity of anti-MDA5 antibody-associated acute interstitial lung disease as a complication of dermatomyositis. Modern rheumatology / the Japan Rheumatism Association. 2011;21:223-7.

C2.  Is clinical course different in ILD subtype (i.e., ARS vs. MDA5 vs. DM vs. PM?)

R2.  Thank you for pointing out an important issues. As mentioned by the reviewer, clinical courses and prognoses of PM/DM/CADM-ILD are heterogeneous. I have described heterogeneity of PM/DM/CADM-ILD in introduction section and section of Clinical phenotype associated with MSAs.

Clinical courses and prognoses of PM/DM/CADM-ILD are heterogeneous. As for underlying diseases, DM-/CADM-associated ILD (DM/CADM-ILD) is more refractory to treatment and has poorer prognosis than PM-associated ILD. (line 45-47).

Recent studies have emphasized the importance of assessment of myositis-specific antibodies (MSAs), which are closely linked to clinical phenotypes of myositis-associated ILD (line 56 – 58). Patients with anti-ARS antibody-positive PM/DM-ILD respond good to glucocorticoid (GCs) treatments and comparatively favorable prognosis. However, GC treatment alone sometimes causes a relapse of ILD (line 88 – 90). In contrast, anti-MDA5 antibody is associated with high incidence of rapidly progressive ILD with respiratory failure and fatal outcome (line 100 – 101). In addition, the presence of anti-MDA5 antibody was associated with increased 90-day mortality and has been recognized as a strong predictor of poor prognosis in patients with PM/DM/CADM-ILD (line 103 -105).

C3.  IIM-ILD: is it different from systemic sclerosis-ILD in regard to pathogenesis and outcome?

R3.  Thank you for pointing out an important issue. As mentioned by the reviewer, comparison of clinical feature and pathogenesis between myositis-associated ILD and systemic sclerosis-associated ILD are areas of great interest; however, this review manuscript is focused on the management of myositis-associated ILD. Therefor, comparison between myositis-associated ILD and systemic sclerosis-associated ILD is beyond the scope of this article.

C4.  A table of poor prognostic factors should be added. 

R4.  Thank you for pointing out. I have added a table of factors associated with poor outcome in PM/CM/CADM-associated interstitial lung diseases in Table 2 (page 10).   

C5.  Table summarizing the treatment efficacy of each drug or drug combination using PICOR (Patients, Intervention, Comparator, Outcome, Results) style should be added. This could show which drug works best for each disease subset.  Is CNI vs antimetabolite in a certain ILD subset?

R5.  Thank you for pointing out important issues. Evidence that supports optimal treatments for patients with PM/DM/CADM-ILD is still very limited, because most of the reports are retrospective nonrandomized observational studies. Although a few prospective therapeutic studies have been recently reported, the evidence levels of each treatment are still not high. I described detailed results of each clinical study in the manuscript. In Table 1, I have just summarized medication, dose, adverse effects and evidence levels.

As suggested by the reviewer, I believe that prospective clinical trial based on stratification of MSA status, especially anti-MDA5 antibody-positive or anti-ARS antibody-positive, is needed to elucidate treatment approaches in various phenotypes of myositis-associated ILD.

Minor comments: 

C6.  "PM/DM/CAMD" should be replaced with "IIM".

R6.  Thank you for pointing out an important issue. Idiopathic inflammatory myopathies (IIMs) includes dermatomyositis (DM), polymyositis (PM), clinically amyopathic dermatomyositis (CAMD), inclusion body myositis, juvenile dermatomyositis, and juvenile myositis; however, ILD are mainly associated with PM or DM or CADM. In this review manuscript, I have mainly described the management of ILD associated with PM/DM/CADM (not inclusion body myositis and juvenile dermatomyositis). For these reasons, I have used “PM/DM/CADM” in this manuscript.

C7.  Page 1, Line 34: "also called myositis" should be removed.

R7.  Thank you for pointing out. As suggested by the reviewer, I have removed them (line 34).

C8.  Page 2, Ln 48. "Acute" vs. "chronic" should be defined more clearly.

R8.  Thank you for pointing out. I have added descriptions in line 48 – 50 as follows:

“The acute/subacute form was defined as progressive ILD with deterioration within 3 months. The chronic form was defined as a slowly progressive ILD presenting with gradual deterioration over a period greater than 3 months. I have added the sentences (Line 48 -50).”

C9.  line 79. "better" than what?

R9.  Thank you for pointing out the error. I have revised it “better” to “good” (line 88).

C10.  Page 3: Table 1: Dose of medication 

R10.  Thank you for pointing out. I described dose of each medication in Table 1.

C11.  Figure 1. "Poor response" should be defined

R11.  Thank you for pointing out. As pointed by the reviewer, the word “Poor response” is vague and unclear. I have used the word “deterioration” and revised in Figure 1 (page 9).

Round 2

Reviewer 3 Report

The author addressed the reviewer's comments adequately.

Author Response

Thank you for so much for reviewing the manuscript. I really appreciate.